# Surprise-Guided Search for Learning Task Specifications from Demonstrations

## Abstract

This paper considers the problem of learning temporal task specifications, e.g. automata and temporal logic, from expert demonstrations. Task specifications are a class of sparse memory augmented rewards with explicit support for temporal and Boolean composition. Three features make learning temporal task specifications difficult: (1) the (countably) infinite number of tasks under consideration; (2) an a-priori ignorance of what memory is needed to encode the task; and (3) the discrete solution space - typically addressed by (brute force) enumeration. To overcome these hurdles, we propose *Demonstration Informed Specification Search (DISS)*: a family of algorithms requiring only *black box* access to a maximum entropy planner and a task sampler from labeled examples. DISS then works by alternating between conjecturing labeled examples to make the provided demonstrations less surprising and sampling tasks consistent with the conjectured labeled examples. We provide a concrete implementation of DISS in the context of tasks described by Deterministic Finite Automata, and show that DISS is able to efficiently identify tasks from only one or two expert demonstrations.

## 1 Introduction

Expert demonstrations provide an accessible and expressive means to informally specify a task, particularly in the context of human-robot interaction. In this work, we study the problem of inferring, from demonstrations, tasks represented by formal *task specifications*, e.g., automata and temporal logic. The study of task specifications is motivated by their ability to (i) encode historical dependencies; (ii) incrementally refine the task via composition, and (iii) be semantically robust to changes in the workspace. However, learning such symbolic specifications is difficult, due to the often combinatorially large search space and lack of gradient-based feedback that can be leveraged. Prior works in this space have used methods ranging from enumeration [20] to mutation-based sampling [11]. In order to more efficiently search for explanatory task specifications, this work introduces a family of approximate algorithms, called *Demonstration Informed Specification Search* (DISS), that systematically reduces the problem of learning from demonstrations into a series of specification identification problems, e.g., finding a DFA that is consistent with a set of example strings [8], a problem more generally referred to as Grammatical Inference [7].

To ground our discussion, we introduce a running example.

**1.1 Running Example** Consider an agent operating in an environment with different regions, which we present as a colored 8x8 discretized world for demonstrative purposes, shown in Fig 1. The agent can attempt to move up, down, left, or right. With probability $1/32$, wind will push the agent down, regardless of the agent's action. The black path is the *prefix* of an episode, in which the agent attempts to move left, slips into a blue region (■), visits a brown region (■), and then proceeds downward.

Given the black demonstration, $\xi_b$, and the *prior* knowledge that the agent's task implies that it will avoid red regions (■), what task, represented as a Deterministic Finite Automaton (DFA), explains the agent's behavior?

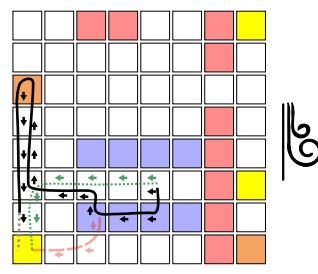

Figure 1

Upon inspecting the demonstration, one might hypothesize that the complete path formed by extending $\xi_b$ with the grey dashed lined to ■ is a positive example of the task. Appealing to Occam's razor, one might conjecture that the task was just to reach ■ and avoid ■. However, under this hypothesis and assuming a temporal discount, $\xi_b$ is quite surprising. For one, the detour to visit ■ seems unjustified. Furthermore, why would the agent not take the red dashed path directly to ■?

To remedy these concerns, one might conjecture that the agent's true task requires visiting ■ after visiting ■ - thus explaining why the agent does not take the red dashed path. Similarly, the demonstration seems less surprising if one assumes that the agent needs to avoid red regions - thus explaining why the agent does not take the light blue dotted path. The result is the task represented by the DFA shown in Fig 2. We shall later systematize this line of reasoning and provide a learner that recovers an explanatory DFA given a demonstration.

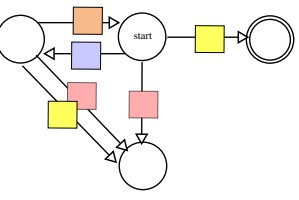

Figure 2

The above example also illustrates a few features that motivate learning task specifications. First and foremost, task specifications directly encode the set of acceptable behaviors and model temporal dependencies. This stands in contrast to Markovian rewards where the encoded task is intimately tied to the details of the dynamics model such as the transition probabilities, time resolution, and workspace configuration [1]. Second, task specifications have well defined compositions, e.g., conjunction and sequential ordering, side stepping many classes of reward bugs stemming from ad-hoc composition [21, 2]. This avoids the need to fine tune learned tasks, which undercuts the original purpose of learning the task representations [13]. Finally, learning task specifications from demonstrations enables learning classes of sparse rewards, a key primitive in sparse-feedback reinforcement learning techniques such as hindsight experience replay [3].

**1.2 Related Work** The problem of learning objectives by observing an expert has a rich and well developed literature dating back to early work on Inverse Optimal Control [10] and more recently via Inverse Reinforcement Learning (IRL) [15]. In IRL, an expert demonstrator optimizes an *unknown* reward function by acting in a stochastic environment. The goal of IRL is to find a reward function that explains the agent's behavior. A fruitful approach has been to cast IRL as a Bayesian inference problem to predict the most probable reward function [16]. To make this inference robust to demonstration/modeling noise, one commonly appeals to the principle of maximum causal entropy [9, 24]. Intuitively, this results in a forecasting model that is no more committed to any given action than the data requires (formalized as bounding the worst-case expected description length of future demonstrations).

While powerful, traditional IRL provides no principled mechanism for composing the resulting reward artifacts and requires the relevant historical features (memory) to be a-priori known. Furthermore, it has been observed that small changes in the workspace, e.g., moving a goal location or perturbing transition probabilities, can change the task encoded by a fixed reward [21, 1].

To address these deficits, recent works have proposed learning Boolean task specifications, e.g. logic or automata, which admit well defined compositions, explicitly encode temporal constraints, and have workspace independent semantics. The development of this literature mirrors the historical path taken in reward based research, with works adapting optimal control [12, 6], Bayesian [17, 23], and maximum entropy [22] IRL approaches.

A key difficulty for the task specification inference from demonstrations literature is how to search an intractably large (often infinite) concept class. In particular, and in contrast to the reward setting, the discrete nature of automata and logic, combined with the assumed *a-priori* ignorance of the relevant memory required to describe the task, makes existing gradient based approaches either intractable or inapplicable. Instead, current literature either (syntactically) enumerates concepts [21, 6, 17, 23] or hill climbs via simple probabilistic (syntactic) mutations [12, 5].

**1.3 Contributions** Specific contributions of our work include:

1. A proxy function whose gradient (i) informs the search for an explanatory task specification; and (ii) is computed with *black-box* access to a maximum entropy planner;
2. A reduction from learning specifications from demonstrations to learning from labeled examples;
3. A guided hill-climbing algorithm that is *agnostic* to the underlying task representation and dynamics model. For example, changing the task representation only requires providing a specification identification algorithm for that class of tasks. Examples include learning decision trees, DFAs, symbolic automata, etc., and
4. An open-source (MIT license) implementation of DISS for learning DFAs that we apply in an empirical setting from prior literature.

The choice of DFAs as the task representation for our experiments was motivated by two main observations. First, DFAs explicitly encode memory, making the contribution of identifying relevant memory more clear. Next, to our knowledge, all other techniques for learning finite path properties from demonstrations focus on syntax defined concept classes. As a result, these existing techniques conflate search efficiency with their concept classes' inductive biases. On the other hand, DFAs constitute a very large and mostly unstructured concept class[1], which allow for learning without user-defined inductive biases.

**1.4 Algorithm Overview** Demonstration Informed Specification Search (DISS) operates by cycling between three components (shown in Fig 3).

1. **Example Buffer**: Given previous iterations, the example buffer yields a set of positive and negative example paths. Based on Simulated Annealing [18].
2. **Candidate Sampler**: Given a set of labeled examples and an optional previous candidate task, the candidate sampler draws a candidate task that is consistent with a set of labeled examples.
3. **Surprisal Guided Sampler (SGS)**: Given task $\varphi$, the SGS algorithm samples a labeled path that is suspected to be mislabeled by $\varphi$.

Figure 3: Overview of DISS.

**Paper Structure** In the sequel, we will formalize the problem statement and agent model based on the extensive literature on maximum causal entropy agent models (Sec 2); (ii) formulate an approximate solution using simulated annealing (Sec 3); (iii) derive a proposal distribution which tries to find mislabeled paths; and (iv) provide empirical evidence for the efficacy of this algorithm for learning DFAs.

# 2 Preliminaries and Problem Statement

**2.1 Dynamics Model** We model the expert *demonstrator* as operating in a *Markov Decision Process* (MDP), $M = (S, A, s_0, P)$, where (i) $S$ denotes a finite set of states, (ii) $A(s)$ denotes the finite set of actions available at state $s \in S$, (iii) $s_0$ is initial state, and (iv) $P(s' \mid a, s)$ is the probability of transitioning from $s$ to $s'$ when applying action $a \in A(s)$. We will make two additional assumptions about $M$. First, we assume a unique (always reachable) sink state, i.e., $P(\$ \mid a, \$) = 1$, denoting "end of episode". Second, we shall assert the Luce choice axiom, which requires that each action, $a \in A(s)$, be *distinct*, i.e., no actions are interchangeable or redundant at a given state [14].

A *path*, $\xi$, is an alternating sequence of states and actions starting with $s_0$: $\xi = s_0 \xrightarrow{a_0} \dots \xrightarrow{a_1} s_n$. Any path, $\xi$, can be (non-uniquely) decomposed into a *prefix*, $\rho$, concatenated with a *suffix*, $\sigma$, denoted $\xi = \rho \cdot \sigma$. We allow $\sigma$ to be of length 0. The last state of $\xi$ is denoted by $\text{last}(\xi) \overset{\text{def}}{=} s_n$. A path is *complete* if it contains $\$$ exactly once, and thus $\text{last}(\xi) = \$$. We denote by $\text{Paths}_\$$ the set of all complete paths, and by Paths the set of all prefixes of $\text{Paths}_\$$, i.e., paths that contain $\$$ at most once.

---

[1]The number of DFAs grows super exponentially in the alphabet size and maximum number of states, e.g., given an alphabet of size four, there are already many many more than 100,000 DFAs with at most 4 states.

**2.2 Task Specifications** A *task specification* (or *task*), $\varphi$, is a subset of paths equipped with a *size* function that measures its description complexity, i.e.,

$$\varphi \subseteq \text{Paths}_\$ \qquad \text{size} : \Phi \to \mathbb{R}_{\geq 0}, \tag{1}$$

where $\Phi$ is a task specifications, called a *representation class*. A *labeled example* is tuple, $x = (\xi, l)$, corresponding to a complete path and a binary label, $l \in \{0, 1\}$. An example, $(\xi, l)$, is consistent with task $\varphi$ if $l = [\xi \in \varphi]$. A collection of labeled examples, $\mathbb{X} = x_1, \ldots, x_n$, is *consistent* with a task, $\varphi$, if they are all consistent.

**Example 2.1.** *Our running example used DFAs over the alphabet $\Sigma = \{\blacksquare, \blacksquare, \blacksquare, \blacksquare, \square\}$ as its representation class. For a DFA task $\varphi$, a path, $\xi$, belongs to $\varphi$ if the corresponding color sequence ends in an accepting (concentric circle) state. For instance, let $\xi_b$ and $\xi_r$ be the completed black and red paths shown in Fig 1 and define $\mathbb{X}_{bg} = \{(\xi_b, 1), (\xi_r, 0)\}$. The DFA shown in 2 is consistent with $\mathbb{X}_{bg}$. We take the size of $\varphi$ to be the number of bits to encode the DFA using stuttering semantics, i.e., default self loops. The concrete encoding is provided by the DFA python library [4].*

Finally, a *candidate sampler* (or *identifier*), is a map from labeled examples, $\mathbb{X}$, and an optional reference task, $\varphi \in \Phi$ to a distribution over consistent tasks in $\Phi$. The lack of a task (either because no reference is provided or no task is consistent, denoted $\bot$. This distribution is denoted $\mathcal{I}(\bullet \mid \varphi, \mathbb{X})$.

**2.3 Policies and Demonstrations** A (history dependent) *policy*, $\pi(a \mid \xi)$, is a distribution over actions, $a$, given a path, $\xi$, where $a \in A(\text{last}(\xi))$. A policy, $\pi$, is $(p, \varphi)$-*competent* if the probability of satisfying $\varphi$ using $\pi$ is $p$, i.e., $\Pr(\xi \in \varphi \mid \pi, M) = p$. A *demonstration*, is a path, $\xi^*$, generated by a employing a policy $\pi$ in an MDP $M$, $\xi \sim (\pi, M)$.

> **Task Inference from Demonstrations Problem** (TIDP): Let $M$, $\Phi$, and $P$ be a fixed MDP, representation class, and task prior, respectively. Further, let $\pi^*$ be a $(p^*, \varphi^*)$-competent policy, $\pi^*$, where $p^*, \varphi^*$, and $\pi^*$ are all unknown. Given a multi-set of i.i.d. demonstrations, $\xi_1^*, \ldots \xi_m^* \sim (\pi^*, M)$, find:
> $$\varphi \in \underset{\psi \in \Phi}{\arg\max} \Pr(\xi_1^*, \ldots \xi_m^* \mid \psi, M) \cdot P(\psi \mid M). \tag{2}$$

By itself, the above formulation is ill-posed as $\Pr(\xi_1^*, \ldots \xi_m^* \mid M, \varphi)$ is left undefined. What remains is to derive a suitable agent model and discuss how to manipulate likelihoods in this model.

**2.4 Task Motivated Agents** Following [22], we propose using the principle of maximum causal entropy to assign a bias-minimizing belief of generating the demonstrations given a candidate task. Here bias-minimizing is taken to mean minimizing the worst case prediction log-loss [24], i.e., the worst-case number of bits needed to encode the the actions of the agent.

We start by defining the causal entropy on arbitrary sequences of random variables. Let $\mathcal{X}_{1:i} \stackrel{\text{def}}{=} \mathcal{X}_1, \ldots, \mathcal{X}_i$ and $\mathcal{Y}_{1:i} \stackrel{\text{def}}{=} \mathcal{Y}_1, \ldots, \mathcal{Y}_i$ denote two sequences of random variables. The *entropy* of $\mathcal{X}_{1:i}$ *causally conditioned* on $\mathcal{Y}_{1:i}$ is:

$$H(\mathcal{X}_{1:i} \mid\mid \mathcal{Y}_{1:i}) \stackrel{\text{def}}{=} \sum_t^i H(\mathcal{X}_i \mid \mathcal{Y}_{1:t}, \mathcal{X}_{1:t-1}) \tag{3}$$

where, $H(\mathcal{X} \mid \mathcal{Y}) \stackrel{\text{def}}{=} \mathbb{E}_{\mathcal{X}}[-\ln \Pr(\mathcal{X} \mid \mathcal{Y})]$, denotes the entropy of $\mathcal{X}$ (statically) conditioned on $\mathcal{Y}$. Intuitively, causal conditioning enforces that past variables do not condition on events in the future. This makes causal entropy particularly well suited for robust forecasting in *sequential* decision making problems, as agents typically cannot observe the future [24].

For MDPs, the unique policy, $\pi_\varphi$, that maximizes entropy subject to a finite horizon and to being $(p, \varphi)$-competent exponentially biases towards higher value actions: $\ln \pi(a \mid \xi) \stackrel{\text{def}}{=} V_\lambda(\xi \cdot a) - V_\lambda(\xi)$. The state and action values are recursively given by the following smoothed Bellman-backup [24]:

$$V_\lambda(\xi) \stackrel{\text{def}}{=} \begin{cases} \lambda \cdot \varphi(\xi) & \text{if } \xi \in \text{Paths}_\$, \\ \text{LSE}_{a \in A(\text{last}(\xi))} V_\lambda(\xi \cdot a) & \text{if } \xi \in \text{Paths} \setminus \text{Paths}_\$, \\ \mathbb{E}_{s'} \left[ V_\lambda(\xi \cdot s') \mid s, a \right] & \text{if } (\xi = x \cdot s \cdot a) \wedge (s, a \in S \times A). \end{cases} \tag{4}$$

Here $\text{LSE}_x \, f(x) \stackrel{\text{def}}{=} \ln \sum_x e^{f(x)}$ and $\lambda$, called the *rationality*, is set such that $\Pr(\xi \in \varphi \mid \pi_\lambda, M) = p$. Unfortunately, $p$ is typically not known. In such cases, the competency of the agent can be treated as a hyper-parameter or estimated empirically, e.g., $p_\varphi \approx {}^1\!/m \sum_{i=1}^m [\xi \in \varphi]$. The former is useful when given on a few demonstrations and the latter is useful when given a large number of demonstrations. Finally, when $\lambda$ is induced from $\varphi$, we shall write $V_\varphi$, and $\pi_\varphi$.

**Explainability of a task** The *surprisal* (or information content) of (i.i.d.) demonstrations, $\xi_1^*, \ldots \xi_m^*$, is the negative log likelihood of the demonstrations under $(\pi, M)$:

$$h(\xi_1^*, \ldots \xi_m^* \mid \pi, M) \stackrel{\text{def}}{=} - \sum_{i=1}^m \ln \Pr(\xi_i \mid \pi, M). \tag{5}$$

Note that the likelihood of i.i.d., demonstrations from $(\pi, M)$ is simply $\exp(-h(\xi_1^*, \ldots \xi_m^*))$. Given a *fixed* MDP, $M$, and a *fixed* collection of demonstrations, $\xi_1, \ldots, \xi_m$, we define the **task surprisal**, $\varphi$, as:

$$h(\varphi) \stackrel{\text{def}}{=} h(\xi_1^*, \ldots \xi_m^* \mid \pi_\varphi, M) \tag{6}$$

Solving a TIDP requires minimizing $h$ plus the negative log prior, which can be taken as $\text{size}(\varphi)$.

# 3 Example Buffer

Given our maximum causal entropy agent model, we employ simulated annealing to approximately solve the TIDP problem. At a high level, *Simulated Annealing* (SA) [18] is a probabilistic optimization method that seeks to minimize an energy function $U : Z \to \mathbb{R} \cup \{\infty\}$. To run SA, one requires three ingredients: (i) a *cooling schedule* which determines a monotonically decreasing sequence of temperatures; (ii) a *proposal* (neighbor) distribution $q(z' \mid z)$; and (iii) a *reset* schedule; which periodically sets the current state, $z_t$, to one of the lowest energy candidates seen so far.

A standard simulated annealing algorithm then operates as follows: (i) An initial $z_0 \in Z$ is selected; (ii) $T_t$ is selected based on the cooling schedule; (iii) A neighbor $z'$ is sampled from $q(\bullet \mid z_t)$; (iv) $z'$ is accepted ($z_{t+1} \leftarrow z'$) with probability:

$$\Pr(\text{accept} \mid z', z_t) = \begin{cases} 1 & \text{if } dU > 0 \\ \min\left\{1, e^{dU/T_t}\right\} & \text{otherwise} \end{cases}, \tag{7}$$

where $dU \stackrel{\text{def}}{=} U(z) - U(z')$; (v) Finally, if a reset is triggered, $z_{t+1}$ is sampled from previous candidates, e.g., uniform on the argmin.

For DISS, we will start by expressing the posterior distribution on tasks in the form:

$$\Pr(\varphi \mid \xi_1^*, \ldots \xi_m^*, M) \propto e^{-U(\varphi)}, \tag{8}$$

where the *energy*, $U$, is given by:

$$U(\varphi) \stackrel{\text{def}}{=} \theta \cdot \text{size}(\varphi) + h(\varphi), \tag{9}$$

and $\theta \in \mathbb{R}$ determines the relative weight of the size. That is, we appeal to Occam's razor and assert that the task distribution is exponentially biased towards simpler tasks, where simplicity is measured by the description length of the task, $\text{size}(\varphi)$, and the description length (i.e. surprisal) of $\xi_1^*, \ldots \xi_m^*$ under $(\pi_\varphi, M)$.

Using the language of SA, we define DISS as follows: (i) $z \in Z$ is a tuple, $(\mathbb{X}, \varphi)$, of labeled examples and a task specification; (ii) $z_0 = (\emptyset, \perp)$; (iii) the proposal distribution, $q(\mathbb{X}', \varphi' \mid \mathbb{X}, \varphi)$ is defined to first sample a concept using an identification map, $\varphi' \sim \mathcal{I}(\mathbb{X})$, then run SGS (defined below) on $\varphi'$ to conjecture a labeled path $\xi$, yielding $\mathbb{X}'' = \mathbb{X} \cup \{(\xi, \xi \notin \varphi')\}$. Next, drop examples from $\mathbb{X}''$ with probability $p_{\text{drop}}$; (iv) resets occur every $\kappa \in \mathbb{N}$ time steps. If a reset is triggered, $\mathbb{X}_{t+1}$, is sampled from $\text{softmin}_{i \leq t} U(\varphi_i)$, and $\varphi_{t+1}$ is sampled from $\mathcal{I}(\bullet \mid \perp, \mathbb{X}_{t+1})$. This process of accepting, rejecting, and resetting defines the example buffer.

# 4 Surprise Guided Sampler (SGS)

The key innovation of DISS is defining a proposal distribution over labeled examples which constrains the concept sampler to find more and more explanatory (less surprising) tasks.

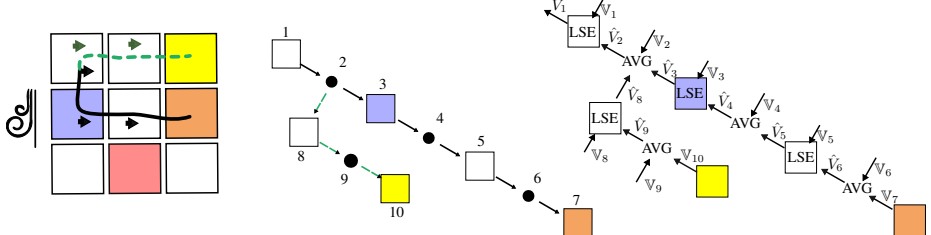

Figure 4: Prefix tree and computation graph with 12 nodes for the paths shown on the left.

We start by discussing the prefix tree of the demonstrations. As we shall see, the prefix tree will serve as a mechanism to reason about the various paths *not* taken.

Let $\xi_1^*, \ldots \xi_m^*$ be a multi-set of demonstrations (paths) and denote by $\mathcal{T} = (N, E)$ the **prefix tree** of the $\xi_1^*, \ldots \xi_m^*$, where $N$ and $E$ are the prefixes (nodes) and edges of $\mathcal{T}$, respectively. Each node $\rho \in N$, corresponds to a prefix of at least one of the demonstrations. Given two prefixes $\rho, \rho' \in N$, $\rho'$ is a descendent of $\rho$ if $\rho' = \rho \cdot y$. An edge connects *parent* $\rho$ to child $\rho'$ if $\rho'$ is the one action (or state) extension of $\rho_i$. For each edge, $(\rho, \rho') \in E$, we define the *edge traversal count*, $\#_{(\rho, \rho')}$, as the number of demonstrations, $\xi^*$, such that $\xi^* = \rho' \cdot y$. A node, $\rho$, is said to be an **ego node** if its prefix ends in a state, i.e. last$(\rho) \in S$. A node that is not an ego node is called an **environment (env) node**. A path, $\xi$, **pivots** at node $\rho$ if $\rho$ is the longest prefix of $\xi$ in $N$. The **pivot actions** (and **pivot states**) of a node, $\rho$, are the set of available actions (states) that result in pivoting at $\rho$, i.e.,

$$A_\rho \overset{\text{def}}{=} \{a \mid \rho \cdot a \in \text{Paths} \setminus N\} \qquad S_\rho \overset{\text{def}}{=} \{s \mid \rho \cdot s \in \text{Paths} \setminus N\}. \tag{10}$$

**Example 4.1.** *Consider the MDP shown in Fig 4 with two paths $\xi_1^*$ and $\xi_2^*$ shown as a green dashed and black solid line resp. The prefix tree of $\{\xi_1^*, \xi_2^*\}$ is shown on in the middle. For convenience an index is associated with each node (prefix). There is a path that pivots at every node except node $\rho_2$, since both possibilities (slipping/not slipping) appear in the demonstrations yielding $S_{\rho_2} = \emptyset$.*

Next, observe that because weighted averaging and LSE are commutative, one can aggregate the values of a set of actions or set of states (environment actions). This motivates defining the **pivot value** of a node $\rho$ as:

$$\mathbb{V}_\rho^\varphi \overset{\text{def}}{=} \begin{cases} \text{LSE}_{a \notin A_\rho} V_\varphi(\rho \cdot a) & \text{if } i \text{ is ego}, \\ \mathbb{E}_s[V_\varphi(\rho \cdot s) \mid \rho, M, s \notin S_\rho] & \text{if } i \text{ is env}, \end{cases} \tag{11}$$

We shall denote by $\mathbb{V}^\varphi \in \mathbb{R}^N$ the node-indexed vector of pivot values associated with task $\varphi$ under our maximum entropy agent model. We note two properties of pivot values. First, pivot values strictly increase as the language of a task specification is made larger:

**Proposition 4.2** (Pivot values respect subsets)**.** *Let $\xi$ be a complete path that pivots at node $i$. If $\varphi \subsetneq \psi$ and $\xi \in \psi \setminus \varphi$, then $\mathbb{V}_i^\varphi < \mathbb{V}_i^\psi$.*

*Proof.* Follows inductively from the monotonicity of $\mathbb{E}$, $\sum$, and $\ln$. □

Second, using the soft Bellman backup (4), one sees that the pivot values **entirely determine** the values, $V$, of the prefixes of the demonstrations (an example is shown on the right of Fig 4). Namely, let $\hat{V}_k(\mathbb{V})$ denote the *derived* value at node $k$ in the prefix tree, and let $\Pr(i \rightsquigarrow k \mid \mathbb{V})$ denote the probability of transitioning from node $i$ to node $k$ under the (local) policy: $e^{\hat{V}_j(\mathbb{V}) - \hat{V}_i(\mathbb{V})}$. This motivates defining the surprisal of the local policy induced by the pivot values as follows:

Let $\mathcal{T} = (N, E)$ be a prefix tree of demonstrations, $\xi_1^*, \ldots \xi_m^*$. The **pivot surprisal** of given $\mathcal{T}$ is map, $\hat{h} : \mathbb{R}^d \to \mathbb{R}$, where $d$ is the number of nodes that can be pivots and:

$$\hat{h}(\mathbb{V}) \overset{\text{def}}{=} -\sum_{(i,j) \in E} \#_{(i,j)} \cdot \ln \Pr(i \rightsquigarrow j \mid \mathbb{V}). \tag{12}$$

Importantly, the task surprisal factors through the pivot surprisal, i.e., $h(\varphi) = \hat{h}(\mathbb{V}^\varphi)$.

**4.1 Pivot surprisal gradients** Motivated by the question "Given a candidate task, what counterfactuals still require explanation?", we ask a related question: "How could the pivot values change to make the demonstrations more likely?" For example, for ego nodes, one might want to make the value of the observed actions large and the pivot value small. The result would be an agent with no incentive to pivot. Unfortunately, changing a pivot value changes the policy in a non-local way, e.g., changing $\mathbb{V}_9$ in Fig 4 also changes the policy for nodes 9, 8, 2, and 1. Fortunately, Prop 4.3 shows that upstream effects are easily summarized by the gradient of $\hat{h}$, with a proof provided in the appendix.

---

**Proposition 4.3** ($\nabla \hat{h}$ determined by local policy). *Letting $p_{xy}(\mathbb{V})$ denote the probability of starting at node $x$ and pivoting at $y$:*

$$\frac{\partial \hat{h}}{\partial \mathbb{V}_k} = \sum_{\substack{(i,j) \in E \\ i \text{ is ego}}} \#_{(i,j)} \cdot \left( p_{ik}(\mathbb{V}) - p_{jk}(\mathbb{V}) \right)$$

$$p_{xy}(\mathbb{V}) \stackrel{\text{def}}{=} \Pr(x \rightsquigarrow y \mid \mathbb{V}) \cdot \left( 1 - \sum_{(y,z) \in E} \Pr(y \rightsquigarrow z \mid \mathbb{V}) \right).$$

(13)

---

Note that Prop 4.3 illustrates that gradients are simple to compute given only access to the policy on the prefix tree. Further, focusing on any given edge one observe that (13) captures the trade-off between (i) making the actions taken more optimal by decreasing the value of other actions; (ii) making the actions taken less risky by increasing the value of possible outcomes.

**Mislabeled counter-factuals** Because of the expressivity of representation classes like DFAs, there is concern that globally optimizing $\hat{h}$ will overfit to the demonstrations and ignore the prior distribution. Thus, our goal is not to simulate gradient descent under $\nabla \hat{h}$, but to instead help identify counter-factual paths that require explanation. Props 4.2 and 4.3 yields the following observation. Let $\xi$ be a complete path with pivot $\rho$ such that: $\xi \in \varphi \iff \frac{\partial \hat{h}}{\partial \mathbb{V}_\rho} > 0$. If $\xi$ is a likely path under $\pi_\varphi$ (and thus has a large effect on $\mathbb{V}$) and the pivot surprisal gradient $\frac{\partial \hat{h}}{\partial \mathbb{V}_\rho}$ is large in absolute value, then $\xi$ may be mislabeled by $\varphi$. For example, if the gradient at pivot $\rho$ is positive, the surprisal and can be decreased by removing path, $\xi$ from the candidate task $\varphi$.

Using these insights we propose surprise guided sampling (Alg 1) which samples a path to relabel based on (i) how likely it is under $\pi_\varphi$ and (ii) the magnitude and sign of the gradient at the corresponding pivot. Combined with an identification algorithm, $\mathcal{I}$, repeated applications of Alg 1 yields an infinite (and stochastic) sequence of tasks resulting from incrementally conjecturing mis-labeled paths.

Importantly, note that Alg 1 only *requires* a black box maximum entropy (MaxEnt) planner to enable assigning edge probabilities, $\Pr(i \rightsquigarrow j \mid \mathbb{V})$, and sampling suffixes given a pivot. If the satisfaction probability of an action is also known, i.e., $\Pr_{\xi'}(\xi \cdot \xi' \in \varphi \mid \xi, M, \pi_\varphi)$, then one can more efficiently sample suffixes using Baye's rule.

---

**Algorithm 1:** Surprise Guided Sampler

**Input:** $\varphi, \mathbb{X}, \mathcal{T}, M, \beta$

Compute $\pi_\varphi$ given $M$ and $\mathcal{T}$.

Let $D = \text{softmax}_\rho \left( -\frac{1}{\beta} \left| \frac{\partial \hat{h}}{\partial \mathbb{V}_\rho} \right| \right)$.

Return $\rho \sim D$ and $\xi \sim (\pi_\varphi, M)$ s.t.

   i $\xi$ pivots at $i$.

   ii $\xi \in \varphi \iff \frac{\partial \hat{h}}{\partial \mathbb{V}_i} > 0$.

   iii $\exists \varphi' \in \Phi$ s.t. $\varphi'$ is consistent with:

$$\mathbb{X} \cup \{(\xi, \xi \notin \varphi)\}.$$

---

## 5 Experiments

In this section, we illustrate the effectiveness of DISS by having it search for a ground truth specification, represented as a DFA, given the expert demonstrations in the workspace from our motivating example (shown in Fig 1). This inference problem is derived from previous benchmarks proposed in [21, 22]. As we expand on below, the key difference is that the representation classes in the prior work are much smaller, i.e., of sizes 930 and 14 tasks respectively. Here we shall work with all DFAs over the alphabet, $\Sigma = \{\blacksquare, \blacksquare, \blacksquare, \blacksquare, \square\}$. As previously discussed, this representation class grows super exponentially, e.g., there are many more than 100,000 DFAs with at most 4 states.

To start, denote the (dotted) green path in Fig 1 that goes directly to 🟨 by $\xi_g$. Simiarly, denote the (solid) black path by $\xi_b$ that immediately slips into 🟪, visits 🟧, then proceeds towards 🟨. This path is incomplete, with a possible extension, $\sigma_b$, shown as a dotted line. The ground truth task is the right DFA in Fig 2. We consider two TIDP instances which vary the representation class and the provided demonstrations. These variants respectively illustrate that (i) the full specification can be learned given unlabeled complete demonstrations; and (ii) our method can be used to incrementally learn specifications from unlabeled incomplete demonstrations. In particular, we will show that DISS supports incorporating prior knowledge about the task, e.g., rules learned from natural language or prior demonstrations.

1. **Monolithic**: $\xi_g$ and $\xi_b\cdot\sigma_b$ are provided as (unlabeled) *complete* demonstrations. The representation class is MinDFA.
2. **Incremental**: $\xi_b$ is provided as an (unlabeled) *incomplete* demonstration. The representation class, $\mathcal{R}$, is a variant of MinDFA that incorporates prior knowledge. Let $\varphi'$ denote the three state DFA for avoiding 🟧 and reaching 🟨. If a task, $\varphi$ is in $\mathcal{R}$, then $\{🟨, 🟨🟨\} \subseteq \text{concept}(\varphi) \subseteq \text{concept}(\varphi')$. That is, prior knowledge is provided that you must reach 🟨, you must avoid 🟧, and you know two positive examples. The size of $\varphi$ is given by: $\text{size}(\varphi) = \text{size}'(\varphi) - \text{size}'(\varphi')$, where $\text{size}'$ is the size function for the MinDFA representation class.

For both experiments, the size weight, $\theta$, was set to $1/50$. For reference, the ground truth task uses about $\approx 40$ nats.

**Candidate Sampler.** To implement $\mathcal{I}$, we adapted an existing SAT-based DFA identification algorithm [19] to enumerate the first 20 consistent DFAs, To make $\mathcal{I}(\varphi' \mid \varphi, \mathbb{X})$ respect the size prior on DFA, we sampled a DFA from the enumerated DFAs, exponentially weighted by the number of bits needed to describe $\varphi'$ given $\varphi$. That is, the sampling was weighted by the change in the number of states and the introduction/removal of labeled edges. Further details on our implementation of $\mathcal{I}$, along with a discussion of other component implementations and their hyperparameters, can be found in the Appendix.

**5.1 Baselines.** As mentioned in the introduction, existing techniques for learning specifications from demonstrations use various *syntactic* concept classes, each with their own inductive biases. Thus, we implemented two DFA-adapted baselines that act as proxies for the enumerative [21, 6, 17, 23] and probabilistic hill climbing style [12, 5] algorithms of existing work:

1. **Prior Guided Enumeration.** This baseline uses the same SAT-based DFA identification algorithm to enumerate DFAs in ordered by the size prior. This is done by finding the $N$ smallest DFAs in lexicographic order (node then edges) as above and then ordering by size. $N = 80$ in the monolithic experiment and $N = 40$ in the incremental experiment. As an alternative to DISS's competency assumption, we allow the enumerative baseline to restrict the search to task specifications that accept the provided demonstrations.
2. **Random Pivot DISS.** As mentioned above, we will evaluate DISS on various SGS temperatures, one of which has $\beta = \infty$. This results in a (labeled example) mutation based search with access to the same class of mutations as DISS, but samples pivots uniformly at random, i.e., no gradient based bias. Note that this ablation still samples suffixes conditioned on the sign of the gradient, and thus the mutations are still partially informed by the surprisal.

**5.2 Results and Analysis** To simplify our analysis, we present time in iterations, i.e., number of sampled DFAs, rather than wall clock time. This is for three reasons. First, for each algorithm, the wall clock-time was dominated by synthesizing maximum entropy planners for each unique DFA discovered, but the choice of planner is ultimately an implementation detail[2]. Second, because many DISS iterations correspond to the same DFAs (due to resets and rejections) the enumeration baseline explored significantly more *unique* DFAs than DISS (a similar effect occurs with the random pivot baseline, since the different pivots give more diverse example sets). This results in the baselines spending more time planning than DISS, thus increasing their time per iterations. Third, the enumeration baseline first enumerates DFAs in lexicographic order (without planning) and then computes the energies in order of increasing size. This incurs a significant ($\approx 15$s) overhead. Thus, using wall clock-time would further skew the results below in DISS's favor.

---

[2]For reference the used for this experiment planner took around 4-10s per DFA.

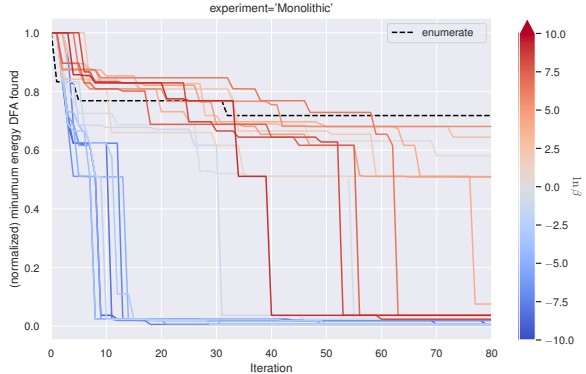
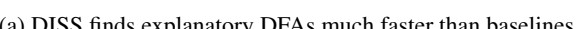

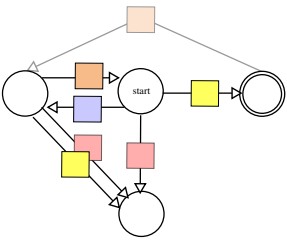

(a) DISS finds explanatory DFAs much faster than baselines.

(b) Most probable DFA found by DISS in the monolithic experiment.

**Search efficiency**  Fig 5a shows the minimum energy of DFA discovered by iteration for the monolithic experiment. For space, the same plot for the incremental experiment is provided in the appendix. To reduce variance, we take the median of 5 runs for each $\beta$. We see that for both experiments, DISS was able to significantly outperform the enumeration baseline (recall that energy is the negative log of the probability) and tended to degrade in its search efficiency as $\beta$ increased. For example, in the incremental setting, $\ln \beta < -5$ typically required only 1-2 iterations (compared to the 13 iterations of enumeration)! For reference, the benchmark this experiment was based on [21] used a syntactic variant of the incremental representation class and evaluated 172 (out of 930) tasks.

The key takeaways are that: (1) DISS is significantly more (cycle) efficient at finding explanatory DFAs than prior based enumeration; (2) Relying on the surprisal gradient (by decreasing the pivot temperature) enables efficient exploration in large concept classes; (3) Using a stronger inductive bias such as asserting partial knowledge of the specification increases the search efficiency of DISS; DISS can be effective even with a few incomplete and unlabeled examples.

**Diversity of DFAs**  In addition to finding the most probable DFAs much faster than the baselines, DISS also found *more* high probability DFAs. The most probable DFA found by DISS for monolothic experiment is shown in Fig 5b. The incremental variant is similar, with an additional edge to the sink failure state enforced by the prior knowledge. We observe that for both experiments, DISS is able to learn that if the agent visits ▪, it needs to visit ▪ before ▪. Nevertheless, our learned DFAs differ from ground truth, particularly when it comes to the acceptance of strings *after* visiting ▪. We note that a large reason for this is that our domain and planning horizon make the left most ▪ effectively act as a sink state. That is, the resulting sequences are effectively indistinguishable, with many even having the exact same energy. In Fig 5b, we make such edges lighter, and note that the remainder of the DFAs show good agreement with the ground truth. This limitation is standard in learning from demonstrations.

# 6   Conclusion

This paper considers the problem of learning history dependent task specifications, e.g. automata and temporal logic, from expert demonstrations. We empirically demonstrate how to efficiently explore intractably large concept classes such as deterministic finite automata to find probable task specifications. The proposed family of algorithms, *Demonstration Informed Specification Search (DISS)*, requires only *black box* access to (i) a Maximum Entropy planner; and (ii) an algorithm for identifying concepts, e.g., automata, from labeled examples. While we showed concrete examples for the efficacy of this approach, several future research directions remain. First and foremost, research into faster and model-free approximations of maximum entropy planners would enable a much larger range of applications and domains. Similarly, while large, the demonstrated concept class was over a small number of pre-defined atomic predicates. Future work thus includes generalizing to large symbolic alphabets and studying more expressive specification formalisms such as register automata, push-down automata, and (synchronous) products of automata.

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
