# Appendix

## 1 Implementation Details

A demo of our implementation of DISS is available at [1]. This implementation resets every 30 iterations, has $p_{\text{drop}} = 1/20$, and uses the cooling schedule: $T_t = 100 \cdot (1 - t/100) + 1$. The provided ipython notebook sweeps through various of SGS temperatures, $\beta \in [2^{-10}, \infty)$. We use the existing maximum causal entropy planner from the prior benchmark [3]. The planning horizon was 15 steps. Because we operate with one or two demonstrations, the rationality, $\lambda$, corresponding to $\pi_\varphi$ is taken to be 10 for all specifications. We also tested a variant with $\lambda$ tuned so that the competency $p$ was close to $4/5$. This provided qualitatively equivalent results, at the expense of inference speed. Finally, two additional inductive biases, which empirically proved necessary for optimizing the baselines, were applied: (i) we remove white tiles, $\square$, from labeled examples; (ii) we transform sequences of repeated colors into a single color thus biasing towards DFA that do not count. For example, $\square\,🟦\,🟦\,\square\,🟨\,🟦 \mapsto 🟦\,🟨\,🟦$.

### 1.1 Candidate Sampler details.

As mentioned in the main text, our implementation of the candidate sampler $\mathcal{I}$ was based on the SAT-based DFA identification algorithm outlined in [2]. Each invocation of the sampler enumerated the first 20 consistent DFAs, lexicographically ordered by the number of states and non-self loops they have. Note that under the lexicographic order it may be the case that a DFA with more states has fewer edges, and thus smaller encoding size (and thus larger prior). This effect is even more pronounced if the reference task, $\varphi'$ is not $\bot$ or the DFA has a sink state, e.g., an irrecoverable failure state. In order to mitigate the blind spot the lexicographic order has for failure sink states, if a subset of symbols, $\Sigma'$, contains only negative examples, a new DFA is created by intersecting the sampled DFA's language with the set of strings not containing $\Sigma'$.

**Example 1.1.** *Suppose* $\mathbb{X} = \{(🟨\,🟥, 1), (🟦\,🟥, 0)\}$. *Then* $\Sigma' = \{🟦\}$ *and the concept sampler will return a DFA that is the intersection with a DFA consistent with* $\mathbb{X}$ *and that always rejects any strings with* 🟦. *Similarly, for* $\mathbb{X} = \{(🟨\,🟥\,🟦, 1), (🟦\,🟥, 0)\}$, $\Sigma' = \emptyset$, *and so the initially sampled DFA is returned unchanged.*

## 2 Incremental Experiment

In the main text, we describe two experiments: monolithic and incremental. For space, the full analysis of the incremental was omitted from the main text. The main purpose of this experiment was to show that (i) the DISS supports changing the representation class; (ii) one can use this to learn a task given prior knowledge; (iii) as one would hope, this prior knowledge accelerates learning. Fig 1a and Fig 1b, illustrate the incremental variants of the energy convergence and learned DFA plots discussed in the main text.

Compared to the monolithic experiment we note that:

1. DISS is able to learn an explanatory DFA within 1-2 iterations for low temperatures.

2. The baselines (enumeration and high temperature pivoting) are now able to find an explanatory DFA; however they still require 10-14 iterations to do so.

Submitted to 36th Conference on Neural Information Processing Systems (NeurIPS 2022). Do not distribute.

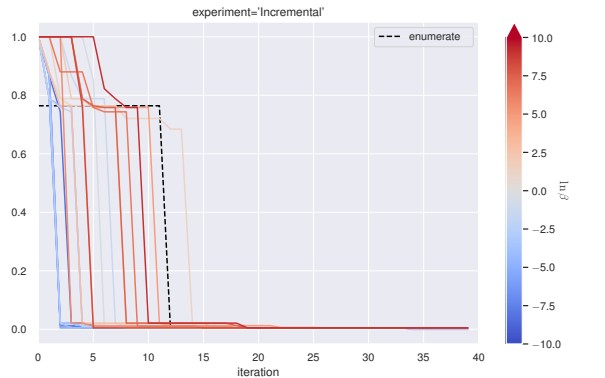

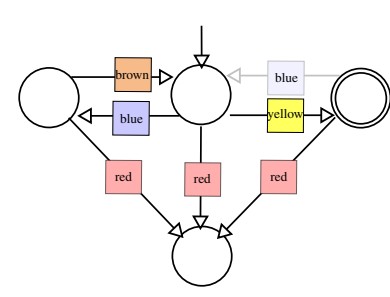

(a) DISS finds explanatory DFAs much faster than baselines.

(b) Most probable DFA found by DISS in the Incremental experiment.

Further, just like the monolithic case, there are many explanatory DFAs that differ in how they continue from the accepting state. Of note is that the prior knowledge now forces all states to transition to the failure sink upon visiting ■, something that did not occur in the monolithic experiment due to lack of evidence. This and the increased convergence show the effects of inductive bias.

## 3   Proof of Prop 4.3

The proof of Prop 4.3 is fairly mechanical and mostly relies on two facts (i) trees only have a single path between any two nodes; and (ii) the gradient of LSE being the soft max distribution. The statement of Prop 4.3 is repeated below as a reminder.

Let $p_{xy}(\mathbb{V})$ denote the probability of starting at node $x$ and pivoting at $y$, i.e.,

$$p_{xy}(\mathbb{V}) \stackrel{\text{def}}{=} \Pr(x \rightsquigarrow y \mid \mathbb{V}) \cdot \left(1 - \sum_{(y,z) \in E} \Pr(y \rightsquigarrow z \mid \mathbb{V})\right) \tag{1}$$

then,

$$\frac{\partial \hat{h}}{\partial \mathbb{V}_k} = \sum_{\substack{(i,j) \in E \\ i \text{ is ego}}} \#_{(i,j)} \cdot \left(p_{ik}(\mathbb{V}) - p_{jk}(\mathbb{V})\right) \tag{2}$$

Our proof will be centered on the following lemma.

**lemma 3.1.** *For any two nodes, $i, k$, in the prefix tree,*

$$\frac{\partial}{\partial \mathbb{V}_k} \hat{V}_i = p_{ik}.$$

*Proof.* Let us first consider the where only a single action or state that pivots at $k$, i.e., $A_k = \emptyset$ or $S_k = \emptyset$. For any edge $(a, b)$, observe that if $a$ is an environment node, then $\Pr(a \rightsquigarrow b \mid \mathbb{V})$ is a constant, denoted $q_{ab}$. Next, observe that because the nodes are arranged as a tree either: (1) $k$ is not reachable from $i$ or (2) only a single edge, call $(i, j)$, can reach $k$ from $i$. Thus,

$$\begin{aligned}
\frac{\partial \hat{V}_i}{\partial \mathbb{V}_k} &\stackrel{\text{def}}{=} \frac{\partial}{\partial \mathbb{V}_k} \sum_{\substack{(a,b) \in E \\ i=a}} q_{ib} \cdot \hat{V}_b(\mathbb{V}) \\
&= \Pr(i \rightsquigarrow j \mid \mathbb{V}) \cdot \begin{cases} 0 & \text{if } \Pr(i \rightsquigarrow k) = 0 \\ \frac{\partial}{\partial \mathbb{V}_k} \hat{V}_j(\mathbb{V}) & \text{otherwise,} \end{cases}
\end{aligned} \tag{3}$$

Similarly, note that because the derivative of LSE is the softmax function, for any ego node $i$,

$$\frac{\partial \hat{V}_i}{\partial \mathbb{V}_k} \stackrel{\text{def}}{=} \frac{\partial}{\partial \mathbb{V}_k} \log \sum_{\substack{(a,b) \in E \\ i=a}} \hat{V}_b(\mathbb{V})$$

$$= \begin{cases} 0 & \text{if } \Pr(i \rightsquigarrow k) = 0 \\ e^{\hat{V}_j(\mathbb{V}) - \hat{V}_i(\mathbb{V})} \cdot \frac{\partial}{\partial \mathbb{V}_k} \hat{V}_j(\mathbb{V}) & \text{otherwise,} \end{cases} \tag{4}$$

where again, $j$ denotes the (potential) unique child of $i$ that can reach $k$. Next, observe that by definition $e^{\hat{V}_j(\mathbb{V}) - \hat{V}_i(\mathbb{V})} = \Pr(i \rightsquigarrow j \mid \mathbb{V})$, using the maximum entropy policy induced by $\mathbb{V}$. Substituting into (4), we see that if $k$ has only a single pivot action (or state) the lemma follows by induction on the path from $i$ to $k$, where the base case is

$$\frac{\partial \hat{V}_k}{\partial_k \mathbb{V}_k} = \left(1 - \sum_{(y,z) \in E} \Pr(y \rightsquigarrow z \mid \mathbb{V})\right) \cdot \frac{\partial}{\partial_k \mathbb{V}_k} \mathbb{V}_k,$$

since probability of applying one of the pivot actions (leading to the forest of subtrees summarized by $\hat{V}_k$) as one minus the probability of traversing an edge in the subtree. $\qquad\square$

*Proof of Prop 4.3.* Recall that the probability of traversing an environment edge is constant w.r.t $\mathbb{V}$. Thus, inspecting (12) we see that it suffices to prove that for any ego edge, $(i, j)$,

$$\frac{\partial}{\partial \mathbb{V}_k} \ln \Pr(i \rightsquigarrow j \mid \mathbb{V}) = p_{ik} - p_{jk}.$$

Recall that by definition, if $i$ is ego, then $\ln \Pr(i \rightsquigarrow j \mid \mathbb{V}) = \hat{V}_j(\mathbb{V}) - \hat{V}_i(\mathbb{V})$. Thus, the proposition follows directly from Lemma 3.1. $\qquad\square$

## 4  KL-Divergence of learned tasks

Below we illustrate that having a lower energy correlates strongly with the induced maximum entropy policy agreeing with the ground truth maximum entropy policy. This agreement is quantified by the expected KL-divergence of the policies at each time step, where the expectation is over the paths produced using the ground truth maximum entropy policy.

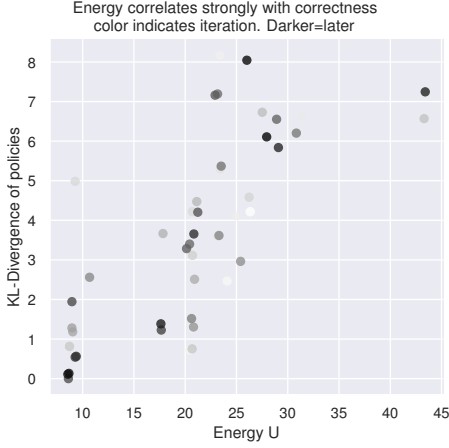

Figure 2: Illustration of KL-divergence of policies