# OpenReview forum: "Surprise-Guided Search for Learning Task Specifications From Demonstrations"
_NeurIPS.cc/2022/Conference — NeurIPS 2022 Submitted_

### Official Review · Reviewer_yBeH · 2022-07-11

**Rating:** 3
**Confidence:** 1
**Soundness:** 2 fair
**Presentation:** 1 poor
**Contribution:** 2 fair

**Summary:**

The authors introduce a method, Demonstration Informed Specification Search (DISS), for learning DFAs from examples. Their method consists of simulated annealing over DFAs and labeled examples simultaneously, with a very complex proposal distribution they call the "surprise guided sampler" (SGS) that involves among other things defining an analogue of "gradient" and incrementally conjecturing paths likely to be mislabeled by candidate DFAs. They compare against a SAT-based algorithm and a more random ablation of DISS on a toy synthetic grid-world domain and find their method performs best.

**Questions:**

Why is this an important problem? Also, could the kernel of the main SGS idea be demonstrated more clearly in a more abstract setting?

**Limitations:**

Yes

**Strengths And Weaknesses:**

The method introduced is very complicated and the toy results are extremely meager. This ratio is highly unfavorable. In general, the paper seems unmotivated. The paper is also highly symbolic and more generally does not seem relevant to the NeurIPS community.

---

> ### Author Response · Authors · 2022-07-31
> **Thanks**
>
> We thank the reviewer for their time and insights. We respond the various questions and concerns as separate comments to support discussion.

---

> ### Author Response · Authors · 2022-07-31
> **Motivation**
>
> This work provides an important step towards solving the inverse problem of non-Markovian variants of reinforcement learning, a topic of growing interest in the community [1, 2, 3].
>
> While we restricted ourselves to gridworld because there existed an *exact* maximum causal entropy planner for such domains, this technique can be applied much more generally.
>
> For example, a stated motivation is to provide a building block for eventually performing hindsight experience replay to accelerate sparse RL learning on complex multi-stages tasks.
>
> [1] Bansal, Suguman et. al. Compositional Reinforcement Learning from Logical Specifications. NeurIPS 2021.
>
> [2] Vaezipoor, Pashootan et. al. LTL2Action: Generalizing LTL Instructions for Multi-Task RL. ICML 2021.
>
> [3] Toro Icarte, Rodrigo et. al. Using Reward Machines for High-Level Task Specification and Decomposition in Reinforcement Learning. PMLR 2018.

---

> ### Author Response · Authors · 2022-07-31
> **General application of SGS idea**
>
> Because the concept identifier and maximum entropy planner are taken black box, our technique can be applied to a range of domains (even continuous) and concept classes (conjunctions of DFAs, reach avoid sets, temporal logic, symbolic automata, etc). We chose to demonstrate SGS in the context of DFAs to emphasize that we do not need to rely on syntactic inductive biases that are present in other concept classes.
>
> The SGS idea itself, however, does leverage the structure of the maximum causal entropy policy which (seems to) restrict its more general transfer outside of learning from demonstrations.

---

### Official Review · Reviewer_Y88L · 2022-07-12

**Rating:** 6
**Confidence:** 3
**Soundness:** 3 good
**Presentation:** 2 fair
**Contribution:** 3 good

**Summary:**

Continuing the tradition of making the powerful goal modeling language LTL amendable to end-user usability, this paper gives an efficient approach to solve for LTL formulas (expressed using a DFA) from expert demonstrations. The crux of their algorithm is incorporating in the fact that the optimal planner is an maximum-entropy planner, using this knowledge to derive an algorithm more efficient than an enumerator agnostic to this fact.


**Questions:**

line53 : where is the light blue dotted path?



**Limitations:**

it is discussed well

**Strengths And Weaknesses:**

## str

This paper shows a promising inference algorithm that is efficient in converting expert-demonstration trajectories into DFA specifications over an MDP environment. Most importantly, it does so by considering fewer numbers of DFAs than prior algorithms. The crux of their algorithm is incorporating in the fact that the optimal planner is an maximum-entropy planner, and leveraging this information to derive an algorithm that is more efficient than enumerating a DFA and using the maximum-entropy planner afterwards to check.

## weakness

### additional baselines are required

This task can be formulated as a straight-forward program synthesis task, and one should seriously consider standard neural-symbolic approaches on it.
Here are few suggestions:
1. For instance, in this simulated environment, one can easily generate paired data of the form (Env, DFA, Traj_from_max_ent_planner) and use supervised learning to learn a mapping from (Env,Traj_from_max_ent_planner)->DFA. One can use hugginface API on a pre-trained language model (gpt-neo) and fine tune it over some structured/textual representation of these abstract objects. Then at inference time simply use the top-k sampling to generate DFA from this fine-tuned language model, and see if it works
2. You can also develop a PCFG capable of generating DFA, then, using a learned neural network, predict the probability distributions of this PCFG, then sample top-k DFA from this grammar, and see if it works

I wouldn't be surprised there will be some very strong patterns in the datapoints (Env, DFA, Traj_from_max_ent_planner) so that a learned neural model can pick it up, and can generate some fairly good (low energy) DFAs. And you won't even have to worry about having a train/test data split since the space of DFA (like you said) is very large, so the likelihood of encountering the same DFA from training to testing is unlikely.

### motivation can be better

Synthesizing a DFA in this manner is very different from synthesizing a DFA for (for instance) regular expression matching. The key difference lies in the evaluation of the DFA, whereas in regex synthesis it is trivial to check if it is consistent with a set of accept/reject input strings, to validate if a DFA is good for specifying a goal, we need to check whether a maximum-entropy planner would produce similar trajectories as the ones given by the expert. This is important and sets this task apart from many other program synthesis tasks, where checking the validity of a program is cheap.

### few writing changes

running example 1.1 is a good attempt (I'm happy to see ML papers are adopting more of a PL paper convention with a running example section), but is still confusing. I still do not know what is the DSL that governs the "goal space" of the robot. For instance, I can infer the robot's trajectory as simply "reach the yellow square as soon as possible, but do so while using as many horizontal movements as vertical movements, while not changing directions too much" <-- is this in the goal space? probably not, as DFA cannot do counting, but how would I know that? In general conter-factuals are easier to describe if the reader knows of the space of hypothesis beforehand. Instead example 1.1 did not explain what this space is, and the reader is lost. Figure 1 has no caption, and Figure 2 has no caption. I have to guess that figure 2 is in fact the solution that explains the behaviour of the robot, i.e. it cannot visit yellow directly after visiting blue, or it will be in the "death" state. A short sentence describe the overall hypothesis space, something like "The goal space of the robot is formulated as a DFA, where edges denote traversing through multiple same-colored blocks (i.e. a blue means visiting multiple blues in succession)...". . . Okay I just took a glance at reference [21], "learning task specification from demonstrations". They explained the problem so much better. The agent moved in the way it moved because it accidentally touched water (due to wind), so it has to dry itself (brown) before touching yellow (electricity), or else it'll be zapped. That is a wonderful story and made much more sense. I don't know why you took their example and removed the most reader-friendly affordance (fire, zap, water, drying) from it. Having to go read that [21] personally only to find out an easier explanation was already there is frustrating at best.

---

> ### Author Response · Authors · 2022-07-31
> **Thanks**
>
> We thank the reviewer for their time and insights. We respond the various questions and concerns as separate comments to support discussion.

---

> ### Author Response · Authors · 2022-07-31
> **Baselines**
>
> The baseline proposed are certainly relevant, however we hesitated to include such baselines for two reasons.
>
> First the distribution of expert demonstrations may not actually match the likely demonstrations of the maxEnt policy. For example, pedagogic demonstrations [1] are often comparatively unlikely, but maximally help the learner disambiguate the objective. Training using the maxEnt policy distribution would fail to take this into account. In fact, one of the stated motivations for DISS listed is to accelerate training neural based approaches to sparse RL by helping define an implicit curriculum that takes into account such ambiguities.
>
> Second, and perhaps most importantly, it's unclear how to create a textual/grammatical representation of the minimized DFA concept class. Straightforward approaches necessarily would favor certain languages by having more syntactic “copies”. One of our main goals in the experiment was to disambiguate syntactic inductive bias from search efficiency.
>
> [1] Ho, Mark K., et al. "Showing versus doing: Teaching by demonstration." Advances in neural information processing systems 29 (2016).

---

> > ### Comment · Reviewer_Y88L · 2022-08-04
> > **I might have messed up my understanding a bit, but baselines are still good**
> >
> > Sorry I got the part about how the DFA and trajectories are related wrong. But I think the rest of my arguments are still solid, the reasons being the following:
> >
> > You are in a completely synthetic environment, with a data-generator that can generate as much synthetic data as possible. It is a very good candidate to even just train a sequence-2-sequence network, mapping one to the other. Unless the dataset of (DFA, Trajectory) is super expensive (np-hard) to construct.
> >
> > So is the dataset of (DFA, Trajectory) expensive to construct? It wasn't quite clear from reading the paper, my understanding is that the mapping from Trajectory->DFA is difficult, which is where your algorithm is supposed to do, but the mapping from DFA->Trajectory shouldn't be too hard?

---

> > > ### Author Response · Authors · 2022-08-04
> > > **Complexity of Dataset**
> > >
> > > We thank the reviewer for engaging with us on this question as it certainly a discussion worth having.
> > >
> > > We agree with the reviewers principle points that:
> > >
> > > 1. Ultimately this is a map from Trajectory to a (list?) of DFAs.
> > > 2. You could try to reduce this to a supervised learning problem.
> > >
> > > The problem (as the reviewer suspected may be the case) is that actually generating a trajectory is computationally non-trivial.
> > >
> > > As is pointed out in [21] even in the (easier) case of non-causal entropy, computing a maximum entropy policy is at least #P-hard (which is harder than NP-hard). [a]
> > >
> > > This could in principle be fine, say if we wanted the most likely paths and that was somehow more tractable. However this becomes a non-trivial problem to overcome when one observes that pedagogic examples are often not the most likely paths [1, 2]. In particular, one may wish to exaggerate a path to emphasize that you were *not* trying to enter the lava. This path would be very out of distribution. Further, environmental stochastic-ness may make the actual demonstration out of distribution. This very event occurs in our motivating example where slipping is what causes the agent to enter the water.
> > >
> > > Finally, and perhaps somewhat surprisingly, it is non-trivial to find "interesting" DFAs for a given environment and enumeration seems to not be sufficient. While there are  many interesting DFAs, they are hard to distinguish from the many more uninteresting (in a given environment) until a policy is actually created.
> > >
> > > [a] To see this, consider a deterministic system. Since all reliable paths are realizable with probability one (due to deterministic actions), the decision of whether to satisfy the task specification can be made first and then policy can select *which* trajectory to sample. Because most paths will not satisfy a complex task specification and the deterministic MDP can be represented as a sequential, this reduces to the problem of uniform sampling from a Boolean circuit. Even sampling a single satisfying path may be non-trivial and is open research problem see [3].
> > >
> > > [1] Ho, Mark K., et al. "Showing versus doing: Teaching by demonstration." Advances in neural information processing systems 29 (2016).
> > >
> > > [2] Dragan, Anca D., Kenton CT Lee, and Siddhartha S. Srinivasa. "Legibility and predictability of robot motion." 2013 8th ACM/IEEE
> > > International Conference on Human-Robot Interaction (HRI). IEEE, 2013.
> > >
> > > [3] LTL2Action: Generalizing LTL Instructions for Multi-Task RL

---

> > > > ### Comment · Reviewer_Y88L · 2022-08-10
> > > > **fair enough, still some ideas that might tickle your fancy**
> > > >
> > > > humans readily generate pragmatic demonstrations, i.e. the #p don't seem to bother us. so simply stating a complexity of a problem (let's be frank here most problems in life are at least np right?) doesn't automatically absolve you from finding a cheap approximate solutions to it. here's something that might be fun to think about:
> > > >
> > > >
> > > > 1. for instance, playing go is probably np-hard if not more, yet you can find a good agent that plays go really well through RL and self-play _alone_. so I don't believe this DFA-traj problem is fundamentally (complexity theory sense) more difficult than playing go. so can we do some way of deriving these approximations ?
> > > >
> > > > 2. sorry to sound like a broken record here, but again having some kind of expertise in human interaction might do good here. you can for instance, give DFA to some participants, and ask them to generate some trajectories showing how the game works. this in itself could be a great dataset, as many folks from cogsci, and communicative acts would be very interested (something similar to mark's construal work recently, but with lava lol).

---

> ### Author Response · Authors · 2022-07-31
> **Motivation for learning Task Specifications (such as DFAs)**
>
> The main motivations for learning task specifications such as DFAs is four fold (and largely overlaps with [21,22]).
>    a) We wish to learn reward representations that are sparse and history based which decouple the set of behaviors we wish to encode from the particulars of the dynamics, e.g., transition probabilities and time scales. This is done by learning sparse-history dependent rewards that explicitly represent the set of “good” behaviors.
>    b) We wish to learn task representations with well behaved compositions, e.g., REGEX operations such as conjunction, disjunctive, and sequencing.
>    c) We wish to learn sparse history dependent rewards as an eventual building block for sparse RL techniques such as hindsight experience replay.
>    d) We wish to learn “auditable” representations of the task, where the representation class (concept identifier) is selected based on the domain and application. For example, DFAs and temporal logic have a rich history of formal verification which could be used to validate downstream synthesis.
>
> With future work, we hope to explore (c) and (d) more exhaustively.

---

> > ### Comment · Reviewer_Y88L · 2022-08-04
> > **a kind of user study would also be nice**
> >
> > for instance you can have some end-users to program "games" such as the ones described in [21], of trying to recharge while having to stay dry at the same time. the user can then demonstrate (pragmatically so) some trajectories, and if your algorithm is good, the agent should be able to perform on novel environments (different worlds) up to satisfaction of the user's judgement.
> > this is bullet proof way of validating a system, but it is expensive, and you run the risk of your DFA not being able to support the "game", for instance, if it involves counting.

---

> > > ### Author Response · Authors · 2022-08-04
> > > **validating system**
> > >
> > > We agree that a user study would strengthen the result.
> > >
> > > That said, there are many practical issues (aside from IRB, funding, etc) including deciding which demonstrations should be shown to the user to get the user to believe that the system works correctly. It may be that the likely trajectories agree with the user's mental model but there is a non-trivial bug. We believe that such a result is beyond the scope of this work and would like to leave as future work.
> > >
> > > That said, inspired by review GoLB, we have included in the appendix an analysis of the KL-divergence of the induced maximum entropy policy from ground truth as a function of the energy. We show that policy agreement strongly correlates with low energy specifications. Furthermore, our initial experiments indicate this holds true for many "reasonable" environment / task /demonstrations pairs, where "reasonable" means pedagogic in the sense of [1,2]
> > >
> > > [1] Ho, Mark K., et al. "Showing versus doing: Teaching by demonstration." Advances in neural information processing systems 29 (2016).
> > >
> > > [2] Dragan, Anca D., Kenton CT Lee, and Siddhartha S. Srinivasa. "Legibility and predictability of robot motion." 2013 8th ACM/IEEE International Conference on Human-Robot Interaction (HRI). IEEE, 2013.

---

> > > > ### Comment · Reviewer_Y88L · 2022-08-10
> > > > **yeah it's hard to run user studies**
> > > >
> > > > the plus side of getting through all that hurdle of a user study is that, it leaves the reviewers with very little room to doubt that your system works. it's kind of a "pick your own poison" scenario. I feel formal methods folks are weary of end-users, partly because these approach tends to work in highly controlled environments and are brittle (i.e. not robust). In a way by having end-users as a gate-keeper of quality, it in a sense forces you think more deeply about whether your approach is ultimately useful and generalizable.
> > > >
> > > > I also agree that perhaps for the scope of this work (after all you the author gets to decide the scope) it is a tad too much.

---

> ### Author Response · Authors · 2022-07-31
> **Proposed Writing Changes**
>
> We sincerely thank the reviewer for their feedback on the presentation - particularly of the motivating examples. This helps answer an internal debate regarding the exposition. As the reviewer writes, the motivating works [21, 22] explicitly assign semantics to the colored tiles!
>
> Our concern was that the reader would bias how they interpreted the demonstrations based on this story and not engage with the counterfactual reasoning. However, we now believe that this is a risk worth taking and that the semantic annotations and story will help guide the reader - as well as provide better grayscale support.
>
> We have access to assets to make this a trivial change.

---

> > ### Comment · Reviewer_Y88L · 2022-08-04
> > **thx.**
> >
> > I think sacrificing accuracy for intuition here is a good move, LTL/DFA is very abstract, and this is your one-shot at making it intuitive

---

> ### Author Response · Authors · 2022-07-31
> **Blue dotted path**
>
> Apologies for the confusion. This was a last minute change to the figure that was not reflected in the text. This will be corrected in the final version.
>
> The blue dotted path should have been a deviation that takes a shortcut through the red (lava) to the yellow (recharge) tile.

---

> > ### Comment · Reviewer_Y88L · 2022-08-04
> > **thx! it makes sense now**
> >
> > .

---

### Official Review · Reviewer_GoLB · 2022-07-12

**Rating:** 6
**Confidence:** 3
**Soundness:** 3 good
**Presentation:** 2 fair
**Contribution:** 3 good

**Summary:**

The paper investigates the interesting problem of learning formal task specifications (i.e., a DFA) from (expert) demonstrations in a MDP. The main motivation is that symbolic structures (DFAs in this case) offer better properties (e.g., non-Markovian goals, goal composition, less sensitive to perturbations in the environment, etc.) compared to previous "inverse control" reward-based approaches like IRL. The main difficulty is the large search space of DFAs (even for the small toy example considered in the paper). The paper proposes the DISS (demonstration informed specification search) as an approximate solution to learning DFAs from demonstration. The key idea is to construct a structured search space over labeled examples and tasks. A knowledge-injected hill-climbing algorithm can operate efficiently in this search space to uncover the task that best "explains" the demonstrations. The key algorithmic contribution in DISS is the "surprise-guided sampler" which constructs labeled examples. Experiments are conducted on a gridworld and show the proposed method outperforming 2 baselines.

----------

UPDATE: I thank the authors for their detailed responses and being open to suggestions. I generally agree with Reviewer Y88L's comments, especially regarding the experimental aspects (baselines, user studies) and overall clarity of the paper. The authors have addressed most reviewer feedback, incorporated a number of reviewer comments into the paper and have included some more empirical data in the Appendix. I'm more positively inclined towards the paper although I think it remains difficult to assess impact with the current set of experiments. I've revised my score upwards to incorporate all the new information.

**Questions:**

  1. What's the definition of $\varphi(\xi)$? (I'm assuming the same as [21].)
  2. What's the correctness / optimality of the final DFAs wrt to ground truth?
  3. Are there any non-grid-world applications or domains in which these methods can be evaluated?
  4. What's the computational complexity of DISS? How does it compare with the baselines?

**Limitations:**

Yes.

**Strengths And Weaknesses:**

Strengths
  + The problem of learning formal representations of goals directly from expert trajectories (without reward functions) is challenging and important. The proposed approach seems to extend the state of the art in this area. The primary contribution of the paper is algorithmic with the introduction of the DISS algorithm for identifying specifications and the SGS component for selecting labeled examples.
  + The algorithmic ideas seem rigorously developed and novel (although I'm not sure I fully understood the details in Section 4.).
  + Experiments show that the method does perform better than reasonably strong baselines (enumeration and mutation-based).

Weaknesses
  - I found the paper to be a bit hard to read (especially compared to the closely related work in [21], which was much clearer). This paper assumes or omits important definitions (e.g., $\varphi(\xi)$, LSE). Combined with the large amount of new notation, interchangeable terminology (e.g., specification vs task) and typos, I found it challenging to grasp the key ideas on my first reading. Overall, I'd suggest this paper be edited to be more self-contained and clearer.
  - The experiments are somewhat limited. I was looking forward to a deeper analysis into the learned DFAs and in particular their correctness. For example, how "close to optimal" are the DFAs (compared to an optimal DFA)? The DFA for a single example is somewhat analyzed but a more detailed "error" analysis would strengthen the paper.
  - Given the use of grid worlds in this paper and previous works [21, 22] and the above concerns, the overall impact of the paper is somewhat unclear to me. (That said, I'm open to revising my score based on the feedback and other reviewers.)

---

> ### Author Response · Authors · 2022-07-31
> **Thanks**
>
> We thank the reviewer for their time and insights. We respond the various questions and concerns as separate comments to support discussion.

---

> ### Author Response · Authors · 2022-07-31
> **Missing definitions**
>
> We apologize for any confusion. Indeed it seems to have been an oversight that $\phi(\xi)$ was not defined. This is shorthand for the indicator [$\xi \in \phi$]. The definition of LSE can be found on 175, which is admittedly less than ideal due to the page break.
>
> We will improve this exposition in future drafts.

---

> ### Author Response · Authors · 2022-07-31
> **Discussion of correctness**
>
> We are happy to provide an additional (empirical) analysis by comparing the KL-divergence of the maxEnt policies induced by the learned DFAs against the maxEnt policy induced by the ground truth DFA across several settings. This comparison will help quantify the “closeness” of a DFA, if the ground truth is not recovered, by showing how different the resulting policy would be from the ground truth. Note that along the demonstration, this will (by design) correlate with the energy.
>
> We will follow up with these results as soon as personal constraints enable us to compute them.

---

> > ### Author Response · Authors · 2022-08-03
> > **See supplementary material**
> >
> > We have added such an analysis to the supplementary material (section 4 of the appendix) and will be working on revising the text accordingly.
> >
> > The key takeaway is that the energy strongly correlates with the expected KL-divergence of the induced maximum entropy policy and the maximum entropy policy of the ground truth task.
> >
> > This means that the low energy DFAs (which are the ones recommended by DISS) are in agreement with the ground truth regarding which actions to take, even outside the demonstrations.

---

> ### Author Response · Authors · 2022-07-31
> **Impact over [21] and [22]**
>
> We again apologize for confusion. The primary contribution of this work is to greatly improve the efficiency and generality of the *search* for explanatory specifications. In particular, both [21] and [22] consider *finite* representation classes.
>
> - In the case of [21] the concept class contains less than 1000 specifications with part of the specification already known (similar to the incremental treatment). Furthermore, the approach uses the principle of maximum entropy, rather than causal entropy, and assumes a uniform prior over all specifications. Our approach generalizes [21] with the algorithm being recovered if the lattice search algorithm in [21] is applied for concept identification.
>
> - In the case of [22] the considered concept class has ~10 tasks. We show that the planner developed in this paper can be employed to learn over significantly  larger concept classes (i.e. the concept classes presented in our work.)
>
> - Our algorithm trivially supports changing the concept class. Our focus on DFAs was to emphasize that we do not need to rely on syntactic inductive biases.

---

> ### Author Response · Authors · 2022-07-31
> **Wider applications and domains**
>
> Because (i) the maxEnt planner and the concept identification algorithm are used black box and (ii) the algorithm only requires detailed analysis of the policy near the demonstrations, it is in theory straightforward to adapt DISS to alternative, potentially even continuous, domains.
>
> That said, the motivation for DISS was for high-level motion planning where grid-like abstractions are a natural formalism - despite the complexities of the underlying dynamics.

---

> ### Author Response · Authors · 2022-07-31
> **Complexity of DISS over baselines**
>
> DISS introduces two types of overhead compared to the baselines:
>
> 1. DISS introduces a constant factor overhead to compute the gradient. This is a single pass over the prefix tree probability just like the energy calculation done in all baselines. Thus, this does not change the (big O) complexity.
>
> 2. The random pivot baseline uses the same suffix sampling as DISS and thus does not differ meaningfully in complexity. The enumerative approach however does not use the suffix sampling and there would be a difference. The complexity would depend on the effort required to sample using the maxEnt policy. If sampling a suffix using the maxEnt policy requires non-trivial time, then this could be a serious concern. Note however that DISS only required a handful of such suffix queries.
>
> Finally as discussed in sibling comment. While the prior distribution makes general comments about complexity difficult, the results of [21] can be recovered as a special case (near deterministic dynamics + lattice concept classes search).

---

### Meta-Review · Area_Chair_dm2g · 2022-08-27

**Recommendation:** Reject
**Confidence:** Less certain

**Metareview:**

The paper presents a new approach for synthesizing automata-based specifications from sample behaviors. In some ways, this is very related to the problem of generating DFAs from examples, but there are important differences related to this planning context that make it more constrained. I think this is an interesting problem and there is a solid contribution in this work that the evaluation clearly demonstrates.
There is significant scope for improvement in the presentation as expressed in many of the comments in the reviews that I think are fixable in a camera ready version of the paper. I think there is a bigger question of fit with the NeurIPS community that is reflected in the low scores that the paper received. The paper reads much more like a CAV paper than a NeurIPS paper, and that might limit its impact in this community.

**Award:**

No

---

### Decision · Program_Chairs · 2022-09-14

Reject